# Vaccination Confidence among Healthcare Workers: Results from Two Anamnestic Questionnaires Adopted in the COVID-19 and Influenza Campaign

**DOI:** 10.3390/vaccines10111835

**Published:** 2022-10-29

**Authors:** Ihab Mansour, Giulia Collatuzzo, Vittoria De Pasquale, Ilenia Mirra, Catalina Ciocan, Alessandro Godono, Enrico Pira, Paolo Boffetta

**Affiliations:** 1Department of Public Health and Pediatrics, University of Turin, 10124 Turin, Italy; 2Department of Medical and Surgical Sciences, University of Bologna, 40126 Bologna, Italy; 3Stony Brook Cancer Center, Stony Brook University, Stony Brook, NY 11794, USA

**Keywords:** vaccines, vaccination, vaccine hesitancy, vaccine acceptance, vaccine confidence, SARS-CoV-2, COVID-19 vaccines, health care workers, influenza

## Abstract

Background: Following the announcement of the development of COVID-19 vaccines, hesitancy about the safety of vaccinations and their side effects have spread, despite having the approval of international drug agencies. The aim of this study was to test the hypothesis that concern about side effects may have led people to fill out the COVID-19 anamnestic vaccine questionnaire with greater attention compared to the similar instrument used for the influenza vaccination. Methods: We analyzed vaccination questionnaires of 218 healthcare workers (HCWs) who underwent both COVID-19 and influenza vaccines in 2020/2021. Outcomes included self-reported allergies, chronic pharmacological treatments, and chronic diseases. We tested the difference in prevalence, analyzed differences using the kappa statistics and concordance correlation, and explored factors associated with differences in reporting. Results: HCWs reported more allergies to substances other than drugs and a higher prevalence of chronic drug intake in the COVID-19 questionnaires than in the influenza ones. Technical staff reported more drug allergies than physicians, and other HCWs reported more outcomes than physicians in the COVID-19 questionnaire. Conclusions: We found that this population of HCWs reported higher conditions during the 2020 COVID-19 vaccination campaign compared to that of the influenza vaccine. The identification of socio-demographic characteristics of the less vaccine-confident HCWs could help in planning targeted interventions to enhance vaccine adherence.

## 1. Introduction

In Europe, a large-scale COVID-19 vaccination campaign began in December 2020 and prioritized healthcare workers (HCWs), among other categories [1].

HCWs are persons employed in acute or long-term healthcare facilities, that have direct contact with patients or patients’ specimens. They are at increased risk of contracting infections and further transmitting them to patients and colleagues [2].

Across the world, substantial efforts have been made to rapidly develop and produce vaccines against SARS-CoV-2. In Europe, novel methods have been employed to increase the speed of vaccine production and development, and the approval of COVID-19 vaccines has been accelerated thanks to the adoption of rapid review procedures, such as a rolling review, a tool to speed up the assessment of vaccine reviewing data from ongoing studies as they become available, and the engagement of a dedicated expert task force by the European Medicines Agency (EMA) [3].

Unfortunately, in Italy and many other countries, the success of the COVID-19 vaccination campaign was hindered by several obstacles, such as organizational issues, and the March 2021 epidemic surge due to the spread of the more transmissible SARS-CoV-2 Alpha variant [4].

The accelerated pace of vaccine development represents a great accomplishment for science, but can also lead to public anxiety and concerns regarding safety issues, leading to vaccine hesitancy [5].

‘Vaccine Hesitancy’ has been defined by the SAGE Working Group on Vaccine Hesitancy as a ‘delay in acceptance or refusal of vaccination despite the availability of vaccination services.’ Vaccine hesitancy is complex and context-specific, varying across time, place, and vaccine. It is influenced by factors such as complacency, convenience, and confidence [6] and, in 2019, the World Health Organization (WHO) identified vaccine hesitancy as a major threat to global health [7].

Moreover, the SAGE Working Group defined confidence as trust in: (1) the effectiveness and safety of vaccines; (2) the system that delivers them, including the reliability and competence of the health services and health professionals and (3) the motivations of the policy-makers who decide on the required vaccines.

Poor, inadequate, or misguided communication can be a problem in any setting.

As a consequence, this has impacted public confidence in vaccines and the vaccines system, leading to increased vaccine hesitancy and refusal [6].

Within this frame, vaccine confidence is central to the success of a vaccination campaign [8].

Furthermore, HCWs played a key role in vaccine promotion and patient guidance from the beginning of the COVID-19 vaccination campaign, and hesitancy among this population likely had a major impact on the adoption of a successful immunization policy.

When looking at the occupational health of HCWs, immunization against vaccine-preventable diseases has multiple benefits: it would protect the workers; help controll the spread of infections, both at the hospital and at population level; prevent frail hospitalized patients from potential worsening of their condition; and maintain healthcare delivery during potential epidemic outbreaks [9]. However, the immunization rates of other vaccinations among HCWs have often been suboptimal, even for highly transmissible infections such as influenza, measles, pertussis, and hepatitis B [10].

Barriers to vaccination against traditional illnesses include concerns about vaccine effectiveness and safety, medical contraindications, and the belief that the disease may be mild, among other reasons [11].

The intention to get vaccinated can be shaped by several factors, including risk perception and fear of side effects.

This fear was amplified with the COVID-19 vaccination campaign, despite vaccines being one of the most controlled drugs on the market, especially concerning side effects, which in many countries have to be reported individually to some Health Authority [12].

On the other hand, the use of common medicines is not likely to pose a problem considering the number of medicines per capita used in Italy [13].

With the spread of COVID-19, mandatory vaccination policies are being increasingly adopted by healthcare institutions and public health authorities; in particular, in Italy, vaccination against COVID-19 became mandatory in April 2021 (D.L. 44—1st April 2021) for HCWs.

Although several studies demonstrated that HCWs showed a positive attitude to the vaccination campaign [14,15,16], the relatively high proportion of HCWs who refused to be vaccinated raised concern and contributed to hesitation among the general population [17].

As is the case in other campaigns, subjects undergoing COVID-19 vaccination in Italy were requested to complete an anamnestic questionnaire [18].

Concern about side effects and general hesitancy regarding the vaccine may have caused those who underwent the procedure, including HCWs, to fill in the COVID-19 questionnaire more carefully than in the case of more common and familiar practices, such as the influenza vaccination. To investigate this hypothesis, the present work replicates another Italian study [19] analyzing and comparing the questionnaires of the 2020/2021 influenza vaccination to that of the first dose of the COVID-19 vaccine in a population of Italian HCWs.

The present study serves as a comparison to the previous study performed in Italy [19], however, this study was conducted on a larger sample of HCW and in a different setting in order to understand whether different variables influenced the reporting of multiple items at the time of filling the questionnaire.

We aimed to compare the anamnestic questionnaires compiled on the occasion of two different vaccines, namely the influenza vaccine and the newly developed COVID-19 vaccine, in a population of Italian HCWs. We focused on discrepancies in reporting allergies, chronic diseases, and chronic use of medications. The discrepancies observed in filling out the questionnaires may be helpful in interpreting the different attitudes of the HCWs towards the two vaccines at the particular historical moment in which they were provided.

## 2. Materials and Methods

The present study enrolled 218 volunteer HCWs, employed in a university trauma center in Turin, Northern Italy, who were offered influenza and COVID-19 vaccines and who accepted to receive both. We used data systematically collected by the Occupational Medicine Unit during the 2020/21 vaccination campaign. A dataset including anamnestic information for the Influenza and the COVID-19 vaccines administered was created.

Influenza vaccination was performed with VaxigripTetra, which provides active immunization against four influenza virus strains (two A subtypes and two B types). This campaign lasted from October 2020 to December 2020.

COVID-19 vaccination was performed with Comirnaty by Pfizer-BioNTech, consisting of two doses, starting from 27 December 2020 to late March 2021, to cover almost all the HCWs.

The 2020/2021 influenza vaccination was offered to employees free of charge through a vaccination campaign, during which information was given through bulletin boards and the hospital website.

The vaccine administrations were performed by the health assistants in the Occupational Medicine department of the Hospital.

The COVID-19 vaccination was offered for free to employees by e-mail and participation was on a voluntary basis (the mandatory measures came into force in a subsequent period).

At the moment of the vaccine administration, anamnestic information (which refers to any known conditions that occurred before vaccine administration) was collected through a standardized questionnaire (at a national level) provided by the hospital, completed independently by the HCW, and then validated by an Occupational Medicine physician.

In the influenza vaccination campaign, a standardized flu questionnaire was used at the time of vaccination to collect information on general health, different conditions, and medications use (Appendix A). A slightly expanded form was used in the COVID-19 campaign (Appendix B). Both questionnaires were designed by the national health authority and were used on a large population basis and were not specifically built for the aim of this study.

Physicians, nurses, social and health care assistants, and health care technicians who had participated in the influenza vaccination campaign of the 2020/2021 season were selected through the acquisition of anamnestic questionnaires, deposited at the Hospital Occupational Risk archive of occupational risk management. We selected HCWs, including physicians, nurses, social and health assistants, and health care technicians, employed at the trauma center in Turin at the time of the vaccination campaign, who had agreed to participate in research projects led by the University Hospital of Turin and for whom anamnestic information on both influenza and COVID-19 vaccination from the 2020/2021 campaign were available. Potential participants were randomly selected from the roster of HCWs who participated in the 2020/2021 influenza vaccination campaign.

Anamnestic questionnaires of the same HCW were collected at the same archive and matched. Trained researchers created a dataset combining the answers to the influenza and the COVID-19 anamnestic questionnaires.

Overall, 218 subjects with available data on both influenza and COVID-19 vaccination, in season 2020/2021, were selected and included in the analysis. Three outcomes were considered: reported prevalence of allergies, reported prevalence of any chronic disease, and reported chronic use of medications and supplements.

Furthermore, it was possible to distinguish the prevalence of specific allergies (to antibiotics, NSAIDs, other medication, and allergies other than medication-related), chronic health conditions (cardiorespiratory, metabolic, kidney, coagulation, immunodeficiency, autoimmune, neurologic, and other diseases), and types of medications (cardiovascular, antimicrobial, immunosuppressant, drugs acting on the nervous system, and other drugs).

### Statistical Analysis

First, we analyzed the distribution of the three outcomes among the study population and characterized them based on the main sociodemographic data (sex, age, and job title). The prevalence of each outcome was compared between the influenza and the COVID-19 questionnaires.

We also analyzed the determinants of self-reported conditions through multivariate logistic regression models, in which the outcome consists of a categorical variable corresponding to the combination of the answers reported on the two questionnaires, with a reference category for concordance between the two, and two additional values: one corresponding to a positive answer on influenza questionnaire and a negative answer on the COVID-19 questionnaire, and the other to the opposite combination, adjusted for sex, age category, and job title.

Kappa statistics were computed to assess the agreement between questionnaires and to assess whether the observed data significantly deviated from perfect concordance, as well as to test whether the proportion of positive answers was higher in the COVID-19 vaccination form than in the influenza vaccination form. Multinomial logistic regressions [20] were conducted to investigate the potential determinants of discordance between the two questionnaires, using concordant answers as the reference category.

The analyses were conducted using the commands *kap, prtest, logistic*, and *mlogit* on the Stata software v. 16 (StataCorp LLC, College Station, TX, USA) [21].

## 3. Results

The analysis included 218 HCWs, corresponding to 436 total questionnaires. Table 1 illustrates the sociodemographic characteristics of the study population. Women represented 51.8% of the population, while the mean age was 47.8 (95% IC = 46.5–49.1). Physicians accounted for half of the population.

None reported severe adverse events to previous vaccines in the influenza questionnaire, while one HCW notified one event in the COVID-19 form (swollen lymph nodes following influenza vaccine). All the HCWs declared that they felt good and did not have a fever at the time of both vaccines.

Table 2 describes the distribution of the outcomes reported by the vaccination questionnaires. The prevalence of declared allergies was higher in the COVID-19 questionnaires than in the influenza questionnaires (33% vs. 24.8%), with a higher proportion of medication-related allergies reported for influenza, and of other types of allergies for the COVID-19 vaccine.

In addition, HCWs reported more frequently in the COVID-19 than in the anti-influenza questionnaires both medication use (40.8% vs. 32.6%) and chronic disease (35.8% vs. 31.7%). The reporting of the three main outcomes varied based on job title, with allergies to medication being more frequently reported by technicians (OR = 7.89, 95% CI 1.11–56.3), other allergies by nurses (OR = 3.03, 95% CI = 1.15–7.94), and use of medication by health assistants (OR = 4.03, 95% CI = 1.26–12.9), than by physicians (not shown in detail).

High concordance was found among the answers for each questionnaire (*p* of difference in prevalence and kappa). Table 2 shows the proportion of HCWs who reported the outcomes in each pairwise combination of questionnaires, and Table 3 illustrates the results of the corresponding multivariate analysis, comparing influenza and COVID-19 questionnaires. No significant associations were found, with the exception of technicians being more likely to report allergies to medication in the COVID-19 form, and subjects 58–66 of age and nurses being more likely to report medication use in the influenza form (OR = 10.0, 95% CI = 1.04–96.4 and OR = 7.31, 1.41–37.9, respectively). No determinants were found for discrepancies in the reporting of chronic diseases.

## 4. Discussion

Our analysis showed that HCWs reported allergies and use of medications more frequently on the COVID-19 vaccination form than on the influenza form. Physicians were generally less prone to declare conditions than other HCWs.

Concern about possible side effects of COVID-19 vaccines began to appear shortly after the announcement of their development, primarily because they were obtained using new technologies. The concern was further amplified during the vaccination campaign. In fact, vaccines are among the most closely monitored medicines. As for other drugs, side effects must be reported to the pharmaco-vigilance authorities [22].

This analysis partially replicates a previous study carried out in Bologna, Italy. To our knowledge, no other studies were conducted with comparable methodology and purpose [8].

The results of this study supported the hypothesis of a tendency for HCWs to over-report anamnestic conditions on the forms for the COVID-19 vaccine compared to the influenza.

Some discrepancies were found between the frequency of drug allergies reported on the COVID-19 vaccine questionnaire compared to the influenza questionnaire by job title (e.g., healthcare technicians reporting more conditions than physicians). This is consistent with the results of the Bologna study [19]. Further multicenter studies with a larger number of participants are needed to identify subgroups of HCWs who are more hesitant towards vaccination. A higher prevalence of prejudice and fear of both influenza and COVID-19 vaccination have been described among nurses compared to other HCWs [23]. Although these cannot be generalized, they suggest higher levels of trust and awareness among physicians, which can be taken into account in future vaccination campaigns in the hospital setting, targeting other HCWs with more aggressive vaccination promotion.

In addition, we observed a higher proportion of drug allergies reported on the COVID-19 questionnaires compared to the influenza questionnaires, coupled with a higher number of drug allergies reported. These results are comparable to those of the previous Italian study [19].

Similarly, allergies to things other than drugs were over-reported in the COVID-19 vaccine questionnaires from the HCWs in this study.

The tendency of over-reporting conditions such as allergies can be interpreted as an aspect of concern for a new drug and may be encouraged by misinformation, even among HCWs [15,24,25].

Moreover, there was also an increase in the prevalence of subjects reporting chronic use of medications, with a concomitant increase in the number of drugs taken per capita. These results confirm the data from the previous Italian study [19].

For both older people and nurses, the OR is increased in both directions (negative COVID positive flu, and positive COVID negative flu), and the confidence intervals largely overlap. It can be hypothesized that these two groups showed a tendency to report discrepant information on drug use, both ways.

These results suggest a mistrust amongst certain workers concerning the COVID-19 vaccine, most likely due to a lack of knowledge on the subject [15,24,25].

These data are in line with from the findings in the literature, in which an association between the same categories of workers and vaccine hesitancy was found [26,27,28]. According to the results of this study, age and gender do not affect vaccination hesitancy, as expected from the results of studies carried out before the start of the vaccination campaign [27,29,30].

Older population groups [23,26,31,32] and nurses [26,27,28] are known to be more hesitant to receive the COVID-19 vaccine. In contrast, among HCWs, older age groups were more likely to be willing to be vaccinated against both influenza and COVID-19 [27,28,30,33].

The different conditions in which medical history was collected should be considered in the interpretation of the results. Indeed, the anamnestic questionnaires concerning the influenza vaccination were administered by specialized nurses, whereas the COVID-19 vaccine questionnaire was completed by the subjects directly. Therefore, although the anti-influenza vaccine questionnaire included less detailed questions on medical history and drug use than the corresponding COVID-19 form, the fact that it was administered by health personnel could have allowed a greater validity and completeness of the answers.

We observed a 4.1% increase in the prevalence of subjects reporting chronic diseases on the anti-COVID-19 vaccine form compared to the anti-influenza vaccine form, in line with the study’s hypothesis, with a concomitant increase in the average number of diseases per capita. Although these differences were not statistically significant, they are in line with those observed in the Bologna study [19]. It is possible that the different instruments used on the two vaccination occasions introduced misclassification and reduced the power of our study.

It is interesting to note that, in this study, gender was not significantly associated with any of the outcomes considered. Data on vaccine hesitancy according to gender are inconsistent, with some studies pointing to a greater attitude in favor of vaccination in men [26,27,28,33,34,35], although some claim the opposite [36,37]. In the previous Italian study, women reported chronic use of medicines more commonly than men on the COVID-19 vaccine questionnaire.

On to the influenza vaccination form, no HCW reported adverse effects secondary to previous vaccinations, whereas one case of lymph adenomegaly was reported as ‘severe’ in the COVID-19 questionnaire, following previous influenza vaccination. However, this type of reaction is considered to be common and has limited pathologic significance.

In this analysis, we described the difference observed in a sample of HCWs undertaking two different vaccines (influenza and COVID-19) in two very close timeframes.

Thus, the differences we observed in the declared health conditions are likely to be due to minor confidence, representing a proxy for the level of COVID-19 vaccine acceptance.

It should also be pointed out that the time of data collection is peculiar, as the anamnestic questionnaires collected date back to different times. The influenza questionnaires were filled out during the last months of 2020, when the vaccination was still the only one available and was recommended not only to frail individuals, but also to HCW, to protect them against this infection. This timing also coincided with the end of the second pandemic wave, a time of general uncertainty [38].

The second questionnaire, on the other hand, was collected at the time of the first SARS-CoV-2 vaccination, in early 2021. At this time, priority was given to HCW and frail individuals. Therefore, a portion of the vaccinable population could be frightened by a newly developed vaccine. Thus, on the one hand, HCW enjoyed the privilege of being vaccinated first, which might have motivated them, but, on the other hand, they might have been frightened by the novelty of the vaccine.

Considered from another perspective, these results may be read to suggest that the higher the confidence in the vaccine, the less accurate the anamnestic reports will be. That is to say, given the higher acceptance and lower hesitancy towards influenza vaccines, HCWs tend to under-report conditions, including allergies and use of medications.

Anamnestic forms are aimed at individuating potential contraindications to the vaccination, distinguishing risk factors of adverse reactions and identifying individuals who could benefit from particular procedures (e.g., adjuvated influenza vaccine in immunodeficient [39] or elderly [40] subjects).

Although the discrepancies found raise concern about the reliability of anamnestic questionnaires, they are a useful tool to collect important information on the general health of the subjects undergoing vaccination, enabling us to identify potential conditions of risk or frailty.

This study consists of some limitations. We analyzed a small sample of HCWs, limiting the generalizability of the results and the possibility of detailed observations by subgroups. In addition, selection bias may have occurred; because we examined COVID-19 vaccination questionnaires of HCWs who had already voluntarily participated in the influenza vaccination campaign, this group could be characterized by a higher propensity to be vaccinated, and consequently less hesitant than other colleagues, particularly as influenza vaccination is not mandatory in Italy, unlike the COVID-19 [32,41] vaccine.

In addition, it should be noted that the COVID-19 vaccination questionnaires were collected in the first months of the campaign when participation was voluntary. This may have led to an underestimation of the hesitancy of HCWs overall as an occupational category, despite representing a valuable population to be addressed with this particular analysis.

An additional limitation of this study is that the observed lower frequency of allergies in influenza vaccine questionnaires could be determined by a previous influenza vaccination.

Another limitation is the partial difference between the two questionnaires; although the questions were comparable, they were not exactly identical, for example, in the COVID-19 vaccine questionnaire, there were more numerous and detailed questions about existing pathologies and the use of drugs. Nevertheless, even when limiting the analysis to the common questions, a higher number of conditions were declared in the COVID-19 forms.

Finally, the two questionnaires were administered to the patients under different circumstances.

## 5. Conclusions

This study showed that this population of HCWs reported a higher prevalence of conditions in the COVID-19 questionnaire than in the influenza questionnaire. Such discrepancies in the anamnestic history reported may be used as a marker of an attitude of suspicion and fear towards a new vaccine, even in a health care setting, which is often caused by the use of unknown technology, and the vaccine being developed in a short time, despite solid evidence of its safety or despite international medicine agencies approval, such as the EMA or FDA. It should be taken into account that this study refers to the very first months of the COVID-19 vaccination campaign, and the first influenza campaign since the COVID-19 pandemic. The HCW included in this study were volunteers for both vaccinations, given that neither influenza nor COVID-19 vaccines were mandatory at the time of the data collection. This may imply that the population we described may have been motivated in being administered the vaccination as soon as it was made available, but were also the first people to face the uncertainty linked to the new vaccination.

We believe that the introduction of mandatory vaccination for health professionals, and the development of vaccination awareness campaigns targeted at the most hesitant categories of workers, are needed. This might improve safety awareness, with a positive influence on other vaccination campaigns.

Based on the results of this study, we believe the investigation of vaccine hesitancy through the comparison of anti-COVID-19 and anti-influenza questionnaires is useful to explore the impact of the pandemic. In addition, the comparison with other vaccination campaigns before the COVID-19 outbreak would be interesting for the improvement of current public health policies. We believe that highlighting socio-demographic characteristics of the less vaccine-confident HCWs could lead to planning targeted interventions, such as conferences and seminars.

We plan to extend this study to questionnaires from the 2021/2022 influenza campaign by reducing the differences in the conditions of collection of the medical history, and subsequently, having an anamnestic interview performed by a physician and standardizing the questionnaires for both vaccines.

## Figures and Tables

**Table 1 vaccines-10-01835-t001:** Distribution of selected characteristic of the study population.

Characteristic	Number (%)
Sex Male Female	105 (48.2) 113 (51.8)
Age 20–40 41–48 49–57 58–66	56 (25.7) 53 (24.3) 51 (23.4) 58 (26.6)
Job Medical doctor Nurse Health assistant Medical technician	109 (50.0) 61 (28.0) 32 (14.7) 16 (7.3)

**Table 2 vaccines-10-01835-t002:** Distribution of outcomes reported by the study population and the respective difference in prevalence and kappa statistics.

Outcomes	Influenza *n* (%)	COVID-19 *n* (%)	Difference in Prevalence	Kappa Coefficient (*p*-Value)
Allergies * Medication-related Other than medication-related	54 (24.8) 34 (15.6) 33 (15.1)	72 (33.0) 39 (17.9) 45 (20.6)	0.06 0.3 0.07	0.59 (<0.001) 0.50 (<0.001) 0.79 (<0.001)
Chronic medication use	71 (32.6)	89 (40.8)	0.03	0.63 (<0.001)
Chronic diseases	69 (31.7)	78 (35.8)	0.18	0.40 (<0.001)
One disease	54 (24.8)	61 (28.0)		
Multiple diseases	15 (6.9)	17 (7.8)		

* Numbers do not sum up to the total because of missing information on the specific allergies.

**Table 3 vaccines-10-01835-t003:** Multiple logistic regression for discordance in reporting the outcomes on COVID-19 and influenza questionnaires (reference category: concordant reports).

Characteristics	Positive at COVID-19, Negative at Influenza OR, 95% CI, *p*-Value	Negative at COVID-19, Positive at Influenza OR, 95% CI, *p*-Value
	**ALLERGIES TO DRUGS**	
Age 20–40 41–48 49–57 58–66	Ref 1.22, 0.07–22.1, 0.895 3.08, 0.27–35.1, 0.365 5.30, 0.42–66.4, 0.196	Ref 0.79, 0.05–13.8, 0.873 0.86, 0.05–15.03, 0.918 1.34, 0.08–23.2, 0.840
Sex Male Female	Ref 2.52, 0.42–14.9, 0.310	Ref 0.69, 0.08–5.72, 0.734
Job Medical doctor Nurse Healthcare assistant Medical technician	Ref 3.18, 0.25–39.9, 0.371 4.45, 0.69–80.6, 0.098 19.6, 1.45–265.5, 0.025	Ref 3.14, 0.61–83.1, 0.117 * *
Pseudo R2 0.16		
	**OTHER ALLERGIES**	
Age 20–40 41–48 49–57 58–66	Ref 2.83, 0.48–16.6, 0.250 1.27, 0.16–9.7, 0.821 1.09, 0.14–8.49, 0.934	Ref 3.74, 0.94–14.9, 0.061 0.87, 0.16–4.67, 0.870 1.96, 0.44–8.81, 0.379
Sex Male Female	Ref 1.02, 0.26–4.00, 0.982	Ref 1.17, 0.43–3.20, 0.759
Job Medical doctor Nurse Healthcare assistant Medical technician	Ref 0.59, 0.12–3.30, 0.549 0.65, 0.07–6.50, 0.714 1.20, 0.13–11.4, 0.872	Ref 2.67, 0.88–8.08, 0.081 2.09, 0.50–8.72, 0.311 1.22, 0.13–11.3, 0.862
Pseudo R2 0.06		
	**CHRONIC DISEASES**	
Age 20–40 41–48 49–57 58–66	Ref 2.01, 0.73–5.52, 0.178 0.75, 0.22–2.55, 0.640 1.15, 0.38–3.51, 0.806	Ref 2.69, 0.61–11.8, 0.189 4.34, 1.09–17.3, 0.038 1.75, 0.37–8.21, 0.475
Sex Male Female	Ref 1.42, 0.62–3.24, 0.404	Ref 1.10, 0.42–2.79, 0.860
Job Medical doctor Nurse Healthcare assistant Medical technician	Ref 1.78, 0.72–4.38, 0.211 1.01, 0.28–3.70, 0.988 1.95, 0.46–8.37, 0.366	Ref 1.48, 0.49–4.42, 0.487 1.65, 0.45–6.13, 0.452 2.64, 0.57–12.2, 0.213
Pseudo R2 0.11		
	**MEDICATIONS OR SUPPLEMENTS USE**	
Age 20–40 41–48 49–57 58–66	Ref 0.93, 0.21–4.06, 0.925 2.01, 0.55–7.37, 0.295 4.22, 1.18–15.1, 0.027	Ref 0.91, 0.05–15.8, 0.948 1.98, 0.16–24.08, 0.591 10.0, 1.04–96.4, 0.046
Sex Male Female	Ref 0.77, 0.31–1.93, 0.578	Ref 0.27, 0.06–1.29, 0.100
Job Medical doctor Nurse Healthcare assistant Medical technician	Ref 3.63, 1.23–10.7, 0.019 2.18, 0.62–7.71, 0.224 2.15, 0.39–11.9, 0.380	Ref 7.31, 1.41–37.9, 0.018 1.31, 0.11–14.6, 0.824 *
Pseudo R2 0.02		

OR, odds ratio, adjusted for sex, age and job title; CI, confidence interval; Ref, reference category; * number of observations too small for the statistical analysis.

## Data Availability

The data presented in this study are available on request from the corresponding author.

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
