# Peer review of "Vaccination Confidence among Healthcare Workers: Results from Two Anamnestic Questionnaires Adopted in the COVID-19 and Influenza Campaign"

_vaccines, 2022, doi:10.3390/vaccines10111835_

Round 1

Reviewer 1 Report

The study compares the information on self-reported allergies, chronic pharmacological treatments, and chronic diseases reported for influenza vs COVID-19 vaccination campaigns by Health Care Workers at a University Hospital in the north of Italy.  The aim of the “study was to test the hypothesis that concern about side effects may have led people to fill out COVID-19 anamnestic questionnaire vaccine with greater attention compared to the similar instrument used for influenza vaccination”.

Introduction

The introduction requires a more in-depth reflection on the literature related to their main topic. Which gap in the literature will this study help to fill? As mentioned by the authors this report replicates the study by Collatuzzo et al (2021). The studies are similar, and some parts are the same. What does this research add to the knowledge in the area? The abstract provides this conclusion: development of campaigns aimed to inform the most hesitant groups of HCWs could improve compliance towards further vaccinations. This conclusion does not derive directly from the results of the study.

A reference is required to support the statement on line 54.

The aim and the methods are inconsistent. The authors did not evaluate concern about the side effects of the vaccines. The questionnaire they applied provided information on allergies, chronic pharmacological treatment, and chronic diseases but no information was obtained on vaccine safety concerns.

Method

How many potential participants were on the HCW roster? How was the random selection process performed?

Statistical analysis. In this section, it is indicated that multiple logistic regression was performed. The authors used STATA software and provided the commands, which are useful. They presented mlogit in the list. This command performs multinomial (polytomous variables) logistic regression. Under this model which were the dependent variable categories analyzed?

 The authors reported Lin’s concordance correlation coefficient. Is this an appropriate statistical test for the type of variables (categorical) analyzed in the study?

Results

In relation to Table 2 the authors mentioned “Also, HCWs reported more frequently in the COVID-19 than in the anti-influenza questionnaires both medication use (40.8% vs 32.6%) and chronic diseases (35.8% vs 151 31.7%)”. However, chronic disease did not have a statistically significant difference between the groups. The p-value for allergies is not provided. Additionally, Table 2 presents CCC and Kappa p-values, but the coefficients of these statistics are not given. This limits the possibility of the reader identifying the level of concordance observed in the study, because only the level of p-value is given, and CCC is a test for continuous variables. Why medication related and no- medication-related allergies not add-up to the total number of personnel with allergies? Why does the table have NA categories?

Table 3 presents the models’ results, apparently, one for COVI19 and the other for influenza. It is not clear how the models were specified. The authors should explain their data analysis in more detail. Additionally, the labels in the tables 3 are not properly positioned, making it difficult for the reader to identify what each column contains. It appears that in the models the dependent variable would be concordance in the reporting and independent variables self-reported allergies, chronic pharmacological treatments, and chronic diseases, and variables such as age, sex, and job, were also included in the model. The authors did not present the main independent variables measures of association (self-reported allergies, chronic pharmacological treatments, and chronic diseases).

The author wrote: “This study showed a tendency among HCWs to report more frequent events and conditions considered associated with COVID-19 vaccine side effects”. It is not clear the use of the term “tendency”; since there is no specific statistical test for tendencies or trends in the results. There is a major issue with this study, in that the authors wish to investigate indicators that would favor vaccine participation, yet everyone in their sample was vaccinated. Would this sample be appropriate for answering the research question? The questionnaires for the study do not include any questions regarding the side effects of vaccines.

This was considered in the study aim. Whether or not there are concerns about COVID-19 vaccine side effects is not necessarily implied by the fact that participants reported allergies more frequently on the COVID-19 questionnaire compared to the influenza questionnaire. This discrepancy may be due to other factors, for example, the participants may have considered COVID-19 to be a new vaccine and wanted to provide more accurate information for future research.

There are several interesting variables available in the questionnaire that were not analyzed, including previous COVID-19 infection, or in the last month have you been in contact with a person infected with SARS-CoV-2?

Reviewer 2 Report

Dear authors, the manuscript is moderately interesting and needs major revisions, to follow some considerations:

- please check if the Keywords are MeshTerms;

- the introduction is too simple and should better explain why this study is important;

- not Vaxgrip Tetra but Vaxigrip Tetra (line 76), also insert bibliography;

- insert bibliography for the Comirnaty vaccine;

- insert bibliography for the questionnaire;

- please explain sampling better;

- Did the authors use multivariable or multivariable logistic regression? I believe it is multivariable.

- the statistical analysis reported in materials and methods is shown roughly, indicate if the Odds ratio, confidence interval, significant p-value etc ... have been calculated all tests performed must be clearly shown in the text and in the tables .

- table 3 is not adequate, reformat it in a simpler way;

-  in materials and methods please indicate the regression for which covariates have been adjusted and indicate the confidence interval used;

- the conclusions are trivial, the authors should better explain how this study should help the policy-maker to improve the situation;

- for the logistic regression it would be advisable to have the "R-squared" (coefficient of determination);

- the discussion must deepen by comparing this study with similar studies, comparing the results with studies for example on the perception of vaccines after adverse events.

- please check english.

Please answer point by point.

Kind regards.

Reviewer 3 Report

In this study the authors tried to compare attitudes towards covid-19 and influenza vaccination in a small sample of HCWs. I have several concerns regarding this paper.

First of all, the manuscript title is misleading since no attitudes towards vaccination were measured. The authors rather compared screening questionnaires administered to subjects before vaccine administration.   Indeed, if a subject came to and filled in the screening questionnaire, he/she was actually seeking for a jab (or in case of the covid-19 vaccination was obliged to). On the other hand, it is highly likely that all included HCWs were willing to receive covid-19 vaccination vaccines, since all were vaccinated against influenza, which is not mandatory. In turn, past influenza vaccination is a very strong predictor of covid-19 vaccination.

Two screening questionnaires are not comparable in terms of closed response options. It is well-known that people tend to select among response options provided in the questionnaire rather than write in the row “Other”

The study did not attempt to control for careless responses. For instance, the observed lower frequency of allergies in flu vaccine questionnaires could be determined by the fact that the past influenza vaccination is the strongest predictor of the current season influenza vaccination. In other words, it is likely that most participants had been already vaccinated against influenza before 2020. They would be therefore more careless in replying. Indeed, a severe allergic episode/Guillen-Barre syndrome following influenza vaccination is the only absolute contraindication to any flu vaccine. I suggest the authors to compare questionnaires before the first and second (and probably third) covid-19 doses and see whether there are differences. Analogously, the sample should be stratified on the basis of previous influenza vaccination, e.g., in two groups, namely (i) vaccinated in 2020 for the first time and (ii) vaccinated in both 2020 and somewhere in the past.      

Round 2

Reviewer 1 Report

Although I had a number of questions about their manuscript, the authors were able to answer them adequately. The article has been significantly improved. The manuscript is recommended for publication.

Reviewer 2 Report

The manuscript is moderately interesting, the authors made the recommended changes.

Kind regards

Reviewer 3 Report

My comments have been generally addressed. I have no further comments.